# The Effectiveness of Herbal Mixture Supplements with and without Clomiphene Citrate in Comparison to Clomiphene Citrate on Serum Antioxidants and Glycemic Biomarkers in Women with Polycystic Ovary Syndrome Willing to Be Pregnant: A Randomized Clinical Trial

**DOI:** 10.3390/biom9060215

**Published:** 2019-06-03

**Authors:** Nava Ainehchi, Arash Khaki, Azizeh Farshbaf-Khalili, Mohamad Hammadeh, Elaheh Ouladsahebmadarek

**Affiliations:** 1Women’s Reproductive Health Research Center, Tabriz University of Medical Sciences, Tabriz 5138665793, Iran; ainehchi.nava@gmail.com; 2Department of Pathology, Tabriz Branch, Islamic Azad University, Tabriz 5157944533, Iran; khaki@iaut.ac.ir; 3Aging Research Institute, Physical Medicine and Rehabilitation Research Centre, Tabriz University of Medical Sciences, Tabriz 5166614766, Iran; farshbafa@tbzmed.ac.ir; 4Department of Obstetrics and Gynecology, University of Saarland, 66421 Homburg, Germany; mehammadeh@yahoo.de

**Keywords:** enzymatic antioxidants, insulin resistance, glycemic index, PCOS, herbal mixture, clomiphene citrate

## Abstract

This paper aimed to evaluate the effect of herbal mixture (*Mentha spicata*, *Zingiber officinale*, *Cinnamomum zeylanicum*, and *Citrus sinensis*) only and along with clomiphene citrate (CC) compared to CC on serum antioxidants, glycemic status, menstrual regulation, and rate of pregnancy. This single-blind randomized clinical trial was carried out on 60 infertile participants with polycystic ovary syndrome (PCOS) willing to be pregnant. They were randomly allocated into group 1 (*n* = 20) who received routine dose of CC pills (50–150 mg) for three menstrual cycles from the fifth day of menstruation for five days; group 2 (*n* = 20) who consumed herbal mixture daily (700 mg); and group 3 (*n* = 20) who used up herbal mixture along with CC for 3 months. Catalase (CAT), glutathione peroxidase (GPx), superoxide dismutase (SOD), malondialdehyde (MDA), fasting blood sugar (FBS), insulin, and homeostatic model assessment for insulin resistance (HOMA-IR) were measured in their blood samples. No statistically significant differences were observed between the three groups in terms of socio-demographic characteristics. After intervention, however, the levels of CAT in group 2 (adjusted mean difference (aMD): = 9.0; confidence interval (CI) 95% = 1.1–16.9) and group 3 (aMD = 12.2; CI 95% = 4.5–19.9), GPx in group 2 (aMD = 986.1; CI 95% = 141.1–1831.1) and group 3 (aMD = 1781.2; CI 95% = 960.7–2601.8), and SOD in group 2 (aMD = 55.1; CI 95% = 26.0–84.2) increased. While FBS in group 3 (aMD = −8.7; CI 95% = −14.7 to −2.7), insulin in group 2 (aMD = −5.6; CI 95% = −10.8 to −0.4), and HOMA-IR in group 2 (aMD = −1.3; CI 95% = −2.4 to −0.2) significantly decreased compared to the group 1. To summarize, herbal mixture supplements along with CC have beneficial effects on serum antioxidant levels, as well as glycemic biomarkers of infertile PCOS, menstrual regulation, and pregnancy rate.

## 1. Introduction

Polycystic ovary syndrome (PCOS) is recognized as a leading cause of infertility with the incidence of 6–26% among women at child bearing age [1]. Clinical manifestations of PCOS as a heterogeneous endocrine disorder are anovulation, hyperandrogenism, amenorrhea or oligomenorrhea, hirsutism, acne, obesity, and dyslipidemia [2]. Polycystic ovary syndrome has been associated with oxidative stress (OS) and metabolic factors, for instance insulin resistance, obesity, and diabetes [3,4]. Oxidative stress can affect female fertility by influencing ovulation, fertilization, embryo development, and implantation [5]. Although reactive oxygen species (ROS) and reactive nitrogen species (RNS) as examples of OS play physiological roles in cellular signaling pathways at low concentrations, they may damage cellular functions in excess levels [6]. Reactive oxygen species are derived from molecular oxygen, and include oxygen ions, free radicals (chemical species with unpaired electrons), and peroxides [3]. Oxidative stress is dramatically increased in PCOS patients, when oxidant/antioxidant status is measured by circulating serum markers, including catalase (CAT), glutathione peroxidase (GPx), superoxide dismutase (SOD), and malondialdehyde (MDA) [3]. However, OS is also correlated with insulin resistance by impairing glucose uptake in muscle and adipose tissue [7]. In vivo and in vitro studies have demonstrated that an excess level of OS results in impaired insulin secretion and insulin resistance [8]. However, antioxidant treatment may ameliorate insulin sensitivity in insulin resistant patients [9]. But whether high levels of OS in PCOS patients derive from PCOS or other potential complications still remains undetermined [10].

Of the most prevalent medications to treat PCOS is clomiphene citrate (CC), long-term consumption of which causes endometrial thickness [11]. Since CC has a structural similarity to estrogen compounds, it binds to estrogen receptors and suppresses the endometrial E2 which plays an important role in endometrial growth and maturation [12]. Therefore, use of an herbal agent with antioxidant and polyphenolic properties without significant side effects helps to treat PCOS as a disease of antioxidant deficiency [13].

*Cinnamomum zeylanicum nees* (Cinnamon) from the Lauraceae family has been known for its antioxidant and anti-inflammatory properties [14]. Cinnamon extract can be used as an antioxidant due to its phenolic content particularly cinnamaldehyde, which improves level of SOD, GPx, CAT, and reduces MDA concentration, as well as increase rate of pregnancy [15,16]. Furthermore, it decreases insulin and blood glucose markedly [17]. *Citrus Sinensis* (L.) Osbeck from Rutaceae family mainly contains hesperidin, polymethoxylated flavonoids (PMF), and terpenoids (limonene and linalool), and these phenolic bioactive compounds indicate considerable cytoprotective effects against OS [18]. Moreover, hesperidin in *C. sinensis* increases the levels of antioxidants including SOD and CAT, and decreases the MDA level [19]. 

Major components of *Zingiber officinale Roscoe* (ginger) from Zingiberaceae family are zingiberene, camphene, and p-cineole, which show antioxidant, anti-cancer, anti-clotting, and anti-inflammatory properties [20,21]. Moreover, it significantly reduces serum levels of fasting blood sugar (FBS) and insulin [22]. In vitro and in vivo studies have confirmed that *ginger* enhances the levels of SOD, CAT, and GP_X_, as well as increasing the antioxidant capacity in blood [23]. *Mentha spicata* (spearmint) from Lamiaceae family, widely spread in the temperate zone, has anti-inflammatory, anti-diabetic, and anticancer features [24]. Some studies have confirmed spearmint contains different volatile compounds such as p-Cymene, isopiperitone, menthone, and β-linalool, and various phenolic phytoconstituents are not only considered an antioxidant source, but they also reduce glucose and OS levels [25,26].

Although aforesaid studies have illustrated the potency of each herb, there are hypotheses that the mixture of these herbs may have more potent antioxidant efficacy. However, to the best of our knowledge, no study has subjected its effect on PCOS. Therefore, the aims of this study were to compare the efficacy of four herbal mixtures with CC on serum antioxidants (CAT, GP_X_, SOD, MDA) and glycemic biomarkers (insulin, insulin resistance, and FBS) as primary outcomes in PCOS patients, and to determine total phenolic content (TPC), total flavonoid content (TFC), free radical scavenging activity, ferric reducing antioxidant potential (FRAP), and phytochemical analysis of herbal mixture as secondary outcome.

## 2. Materials and Methods 

### 2.1. Medicinal Plant and Capsule Preparation 

The dried plant samples consisting of the leaves of *M. spicata*, rhizomes of *Z. officinale*, bark of *C. zeylanicum*, and peels of *C. sinensis* were provided from the Herbal Medicine Market and identified at the Department of Pharmacognosy, Faculty of Pharmacy, Tabriz University of Medical Sciences, Tabriz, Iran. The different plant samples were thoroughly powdered and sieved. The powders were mixed with 5 (250 mg): 4 (200 mg): 3 (150 mg): 2 (100 mg) weight ratios of spearmint, ginger, cinnamon, and *C. sinensis*, respectively. Finally, obtained powder was used for preparation of 700 mg capsules.

### 2.2. Standardization of the Herbal Mixture

#### 2.2.1. Preparation of Methanolic Extract

The methanolic extract was obtained from 72 g of herbal mixture powder using the maceration method with 700 mL methanol (MeOH) during three consecutive days. The obtained extract was filtered and dried using a rotary evaporator at 45 °C. The yielded methanolic extract was 12.04 g.

#### 2.2.2. Free Radical Scavenging Assay

Free radical scavenging activity of methanolic extract was measured using DPPH (2, 2-diphenyl-1-picrylhydrazyl) method. The extract was dissolved in methanol to prepare a stock solution with the concentration of 10 mg/10 mL. Serial dilutions (0.5, 0.25, 0.125, 0.625, 0.313, and 0.156 mg/mL) were made to reduce the concentrations. Two mL of diluted solution was mixed with 2 mL of 0.08 mg/mL DPPH solution and was allowed to stay for 30 min for any reaction. The ultraviolet (UV)-visible absorbance was set at 517 nm by Thermo Fisher spectrophotometer (Walthman, MA, USA). The reduction percentage was recorded against the extract concentration to compute RC50 values (the concentration of extract that provides 50% loss of DPPH activity). Quercetin was used as the positive control and the experiment was performed in triplicate [27].

#### 2.2.3. Ferric Reducing Antioxidant Potential 

Total antioxidant activity was determined by the ferric reducing antioxidant potential (FRAP) following the method of Benzie and Strain with some modifications. This method is based on the reduction of Fe (III)-TPTZ (ferric 2,4,6-tripyridyl-s triazine complex) by antioxidants to Fe (II)-TPTZ (ferrous form). Light blue reagent clarifies the presence of Fe (III)–TPTZ, and changing it to dark blue after interaction with antioxidants illustrates the presence of Fe (II)–TPTZ in the reagent. To this end, FRAP stock solution was prepared freshly by mixing TPTZ (10 mM), HCl (40 mM), and FeCl3 (20 mM) in an acetate buffer (300 mM, pH 3.6). Next, 900 µL of FRAP reagent was allowed to react with 30 µL of plant extract for 30 min in a test tube. The tube was vortexed and placed in bain-marie to reach 37 °C. The standard calibration curve was obtained by using different concentrations of FeSO_4_ as the standard for calculation of the FRAP values for both the quercetin and herbal mixture. Afterwards, absorbance at 595 nm wavelength was monitored against control solution [28]. Finally, the result was expressed in µM Fe (II)/g dry mass and compared with that of quercetin as the standard antioxidant.

#### 2.2.4. Total Phenolic Content

Total amount of the phenolic components was determined by Folin-Ciocalteau method. One mL of methanolic extract (5 mg/mL in acetone-water solution) was mixed with 200 µL of Folin-Ciocalteau reagent and 1 mL of 2% Na_2_CO_3_, and the new solution was incubated for 30 min at room temperature. Then the absorbance was determined at 750 nm using Thermo Fisher spectrophotometer. Various concentrations of gallic acid were utilized as the standard, and control samples did not contain any extract. All evaluations were performed in triplicate.

#### 2.2.5. Total Flavonoid Content

Total flavonoids were determined using the AlCl_3_ method. Eighty percent methanol was used as the test solution. With this end in view, 133 mg crystalline AlCl_3_ and 400 mg crystalline NaCOOCH3 were dissolved in 100 mL of 80% methanol and used as an AlCl_3_ reagent. To estimate the flavonoid content of extract, 2 mL of extract solution, 400 µL of water, and 1 mL of AlCl_3_ reagent were mixed and the absorbance was set at 430 nm using Thermo Fisher spectrophotometer. The blank solution containing no AlCl_3_ reagent and different concentrations of quercetin were used as the standard. The number of flavonoids was estimated based on the calibration curve of quercetin. All measurements were performed in triplicate.

#### 2.2.6. Essential Oil Extraction

The herbal mixture powder (120 g) was exposed to hydrodistillation using a Clevenger type apparatus for about 4 h. Then, the obtained dark yellow oil was dried over anhydrous sodium sulfate, measured, and stored in a dark glass at 4 °C for further analyses. The essential oil (0.5% *w*/*w*) was assessed on the dry weight basis.

#### 2.2.7. Gas Chromatography–Mass Spectrometry and Gas Chromatography–Flame Ionization Detector Analyses

Gas chromatography–mass spectrometry (GC–MS) and gas chromatography with flame ionization detector (GC–FID) analyses were performed on a Shimadzu GC-MSQP-5050A and GC-17A equipped with a DB-1 fused silica column (60 m × 0.25 mm i.d., 0.25 µm film thickness), with an oven temperature of 50 °C rising to 260 °C at a rate of 3 °C/min. The total running time for a sample was about 82 min. Helium was used as the carrier gas at a flow rate of 1.3 mL/min. The essential oil was diluted 1:100 in *n*-hexane and 1 µL was injected into the column. Split ratio, ionization energy, scan time, and acquisition mass range were 1:33, 70 eV, 1 s, and 30–600 amu, respectively.

#### 2.2.8. Identification of Components

Identification of the essential oil components was based on the comparison of the standard alkanes (C8–C20) from Sigma-Aldrich (St. Louis, MO, USA) with the retention times and mass spectral data, and computer matching with the NIST 21, NIST 107, and WILEY 229 library by comparing the fragmentation patterns of the mass spectra with those reported in the library [29].

### 2.3. Clinical Trial Study

This single-blind, parallel randomized clinical trial was authorized by the Ethics Committee of Tabriz University of Medical Sciences (code: TBZMED.REC.1394.576) on 26 November 2015, and was registered in the Iranian Registry of Clinical Trials (IRCT201509295563N7) on 9 January 2016. A total of 60 women with PCOS aged 18–35 years old with primary and secondary infertility, a body mass index (BMI) between 26.5 and 28.5 kg/m^2^, and willing to be pregnant were included in the study. The participants were from the Infertility Clinic, Alzahra Hospital, Tabriz, Iran, and contributed to the study for approximately nine months (from 16 January 2016 to 25 October 2016). The diagnosis of PCOS was based on Rotterdam criteria (2003), satisfying at least two out of three of the following criteria: oligomenorrhea /or amenorrhea, clinical /or biochemical sign of hyperandrogenism, and presence of PCOS by ultrasonography [30]. The criteria for inclusion were PCOS women diagnosed with primary or secondary infertility, aged between 18 and 35 years, and having a BMI < 30. The exclusion criteria included the patients with diabetes mellitus, the use of medications such as those helping ovulation or insulin sensitizers, thyroid disorders, cholesterol-lowering drugs, smoking, current treatment of infertility, hypertension, cardiovascular diseases, Cushing syndrome, and allergy to spearmint, ginger, cinnamon, and *C. sinensis.*

The sample size based on:$$n=\frac{{\left(z_{\alpha }+z_{\beta }\right)}^{2}\left(p_{1}\left(1-p_{1}\right)+p_{2}\left(1-p_{2}\right)\right)}{{\left(\varepsilon -\delta \right)}^{\;2}}$$
95% confidence interval, 80% power, δ=0.5, ε=0.1, and α = 0.05 was calculated 20 individuals per group. Final sample was estimated 25 women, considering 25% probable drop-out for each group.

All participants were informed about the purpose of the study, and an informed consent was obtained from each of them. They were also asked to keep their daily intake of food during three months of study. Compliance of the subjects was followed up via phone consultation every week. The subjects were randomly allocated into the three groups: group 1 (*n* = 20) received routine dose of CC pills (50–150 mg) for three menstrual cycles from the fifth day of menstruation for five days; group 2 (*n* = 20) consumed 700 mg herbal mixture capsule daily; and group 3 used up 700 mg herbal mixture capsule along with CC for three months.

#### 2.3.1. Randomization

The participants were divided into three groups by random allocation software (RAS/ version 1.0.0, M Saghaei, Isfahan, Iran) [31] through randomized blocks of three and six with an allocation ratio of 1:1:1 by a person who was not involved in the study. For allocation concealment, according to sequence generation opaque and sealed envelopes numbered from 1 to 75; each contain a letter designating the allocation. The first envelope was dedicated to first participant and this process was followed to the end of the research. Only the statistician was blind to the study. 

#### 2.3.2. Collection of Serum Samples

Ten mL of blood samples was collected twice from antecubital veins in the morning after an overnight fasting, on the second day of the women’s menstrual cycle; first, as the pre-intervention, and second, as the post intervention (three months later). The blood samples were centrifuged for 10 min at 4000 rpm to separate the serum. The extracted serum was divided into four aliquots and kept frozen at −70 °C until assay.

#### 2.3.3. Measurement of Biomarkers of Oxidative Stress 

Serum MDA levels were measured by thiobarbituric acid (TBA) test with ±0.01 µMol/L sensitivity. Concentration of plasma MDA was determined with a spectrophotometric detector at 532 nm. A 1,1′,3,3′-tetramethoxypropane was used to construct the calibration curve as the standard [32] (reference value: 0.54–1.32 pg/mL). In addition, serum SOD levels were measured by colorimetric method using an ELISA kit (RANDOX, Antrim, North Ireland UK) according to the manufacturers’ instruction. Sensitivity of the assay was ± 0.01 IU/mL (Cat. No. (SD) = 124; reference value: 164–240 IU/mL). Moreover, serum GPx levels were measured by UV method using the ELISA kit (RANDOX). Sensitivity of this assay was ± 1.15 IU/mL (Cat. No. RS 2318; reference value: 4171–10881 IU/mL). Furthermore, serum CAT levels were measured by immunoturbimetric assay using an ELISA commercial kit according to the manufacturers’ protocol (CUSABIO Kit, WUHAN HUAMEI BIOTECH Co., Ltd. Wuhan, China). Sensitivity of the assays was ± 3.9 pg/mL (Cat. No. CSB-E13635h; reference value: 19.8–66.4 pg/mL).

#### 2.3.4. Measurement of Glycemic Biomarkers

Insulin concentration was determined through a fully automated chemiluminescence assay (LIAISON C-Peptide, Byk-Sangtec; reference value: 3.21–16.32 µIU/mL) [33]. Blood glucose level (FBS) was measured by enzymatic methods using commercial kits (Pars Azemun, Isfahan, Iran) and the auto-analyzer system (Selectra E, Vitalab, Netherlands; reference value: 70–115 mg/dL). Homeostatic model assessment for insulin resistance (HOMA-IR) was calculated according to $\mathrm{HOMA}-\mathrm{IR}=\frac{\mathrm{Glucose}\;\times \;\mathrm{Insulin}}{405}$ formula. It is noteworthy to mention that the 75th percentiles of HOMA-IR in the whole population, in normal-weight, and in obese people are 3.027, 1.68, and 3.42, respectively.

#### 2.3.5. Ultrasonography

Volume of ovary, numbers as well as size of follicles were probed by vaginal ultrasound (5 MHz Ulramark 4 Plus; Advanced Technology Laboratories, Bothell, WA, USA).

#### 2.3.6. Checklist of Side Effects

The participants were also asked to complete a checklist encompassing the side effects of medications during the intervention.

### 2.4. Statistical Analysis

Normality of all quantitative variables for each of the groups was confirmed using the Kolmogorov-Smirnov test. Descriptive statistics, including the frequency and percentage, and measures of central tendency and dispersion, including the mean and standard deviation (SD), were also used to describe the study variable. Moreover, the results were analyzed using one-way analysis of variance (ANOVA) with a post-hoc Tukey test for baseline quantitative variables. The paired t-test was also applied for comparison of quantitative data before and after the trial within the 3 groups, and analysis of covariance (ANCOVA) was used for between-group analysis after intervention adjusted for baseline values. *p*-Value ≤ 0.05 was considered statistically significant. The data were analyzed by SPSS software, version 22.0 (SPSS Inc., Chicago, IL, USA).

## 3. Results

### 3.1. Preparation and Analysis of the Herbal Mixture 

In this study, the total methanolic extract was obtained from the herbal mixture powder which was 12.04 g. Then, the levels of DPPH, FRAP, TPC, and TFC were measured from this methanolic extract.

Herbal mixture showed potent antioxidant activity (RC50: 0.018 ± 0.0007 mg/mL) in comparison to quercetin (RC50: 0.004 ± 0.0001 mg/mL) as the positive control; this activity could be attributed to the presence of phenolic structures in the extract. The methanolic extract of herbal mixture had the ability to decrease TPRZ-Fe (III) to TPTZ-Fe (II), and FRAP values for methanolic extract of herbal mixture was 720 ± 35 µmol Fe (II)/g, which was lower than that of quercetin (2880 ± 41 µmol Fe (II)/g) as the standard antioxidant. In addition, the assessment of TPC and TFC contents proved the presence of 24.062 ± 0.2 mg gallic acid/100 mg and 8.93 ± 0.09 mg quercetin/100 mg in herbal mixture (Table 1).

The amount of extracted essential oil was 0.5% *w*/*w,* which was assessed on the dry weight basis. The GC–MS and GC–FID analysis of the herbal powder essential oil indicated that the phytochemicals of the essential oil, including sesquiterpenes and derivatives (55.49%), aldehydes (19.38%), and monoterpenes (15.45%) were the main groups of oil components. Moreover, it was shown that zingiberene (13.58%), α-curcumene (10.77%), β-sesquiphellandrene (9.86%), α-farnesene (5.30%), and β-bisabolene (5.93%), isolated from ginger with sesquiterpenes hydrocarbons, were the most abundant components in essential oil of herbal mixture powder. In the same way, cinnamic aldehyde (13.06%) was separated from cinnamon with the organic compounds of aldehyde, and pulegone (5.61%) was extracted from spearmint with monoterpene structure. No substance was extracted that could be related to *C. sinensis,* which might be due to its lowest dosage (Table 2, Figure 1).

Based on our results, the antioxidant activity increased proportionally to the levels of polyphenol component, and a tremendous effect was seen on the treatment of PCOS regarding antioxidant activity. Therefore, in standardization of herbal mixture, different proportions of each herbal powder were mixed, and phenolic and flavonoid contents were measured. Moreover, the highest levels of phenolic and flavonoid contents were obtained in the weight ratios of 5 (250 mg) spearmint): 4 (200 mg) ginger: 3 (150 mg) cinnamon: 2 (100 mg) *C. sinensis* in comparison with other ratios.

### 3.2. Clinical Trial Data

In the present study, the initial sample consisted of 90 women with PCOS, from whom 11 were excluded because they did not meet the Rotterdam criteria, and 4 women’s husbands suffered from azoospermia. Then, 75 participants were randomly allocated into 3 groups including group 1: CC (*n* = 25); group 2: herbal mixture (*n* = 25); and group 3: CC with herbal mixture (*n* = 25). Twenty-two women out of 25 in group 1, 23 out of 25 in group 2, and 24 out of 25 in group 3 received allocated interventions. However, 3, 2, and 1 of them in groups 1, 2, and 3, respectively, did not feel comfortable enough to participate. During the follow-up stage, 2 participants in group 1 were lost to be followed up with because of consuming other medication along with treatment, 3 of them in group 2 discontinued intervention on account of deciding to get treated with intrauterine insemination (IUI) or in vitro fertilization (IVF), and 2 of them in group 3 did not take an initial blood test, while 2 of them wanted to be treated with IUI or IVF. Finally, 60 participants (*n* = 20 in each group) finalized the intervention and their blood samples were analyzed at the end of three months (Figure 2).

Before intervention, there were no significant differences between groups in terms of socio-demographic characteristics. The mean (SD) age was 25.7(4.1) years; the mean (SD) marital age was 20.2 (3.5) years, and the mean (SD) infertility history was 3.9 (3.0) years. Likewise, the mean (SD) weight was 73.8 (11.4) kg, and the mean (SD) BMI was 27.5 (3.8) kg/m^2^. Furthermore, the numbers of women with primary and secondary infertility were 53 (83.3%) and 7 (16.7%), respectively. None of the participants were smokers (Table 3). 

Before intervention, there were no significant differences between groups regarding the serum levels of CAT (*p* = 0.725), GPx (*p* = 0.751), SOD (*p* = 0.793), and MDA (*p* = 0.835). After intervention, however, with adjusting baseline values, significant differences were observed between groups in terms of CAT (*p* = 0.001), GPx (*p* < 0.001), and SOD (*p* < 0.001) levels except for MDA (*p* = 0.420). In a binary comparison, the CAT levels in group 2 (aMD = 9.0; CI 95% = 1.1–16.9, *p* = 0.021) and group 3 (aMD = 12.2; CI 95% = 4.5–19.9, *p* < 0.001), the GPx levels in group 2 (aMD = 986.1; CI 95% = 141.1–1831.1, *p* = 0.017) and group 3 (aMD = 1781.2; CI 95% = 960.7–2601.8, *p* < 0.001), and the SOD levels in group 2 (aMD = 55.1; CI 95% = 26.0–84.2, *p* < 0.001) and group 3 (aMD = 55.9; CI 95% = 27.5–84.2, *p* < 0.001) significantly increased in comparison with group 1. However, the binary comparison displayed no significant decrease in the MDA levels in group 2 (aMD = −0.1; CI 95% = −0.4 to 0.1, *p* = 0.502) and group 3 (aMD = −0.1; CI 95% = −0.4 to 0.1, *p* = 0.725). In addition, the binary comparison indicated no significant difference in the SOD, GPx, CAT, and MDA levels between groups 2 and 3 (*p* > 0.05).

Yet, in within-group analysis, a significant increase was observed after intervention compared to baseline considering the serum levels of CAT in group 2 (mean difference (MD) = 8.6; CI 95% = 1.9–15.2, *p* = 0.014) and group 3 (MD = 12.8; CI 95% = 6.5–19.1, *p* < 0.001), GPx in group 2 (MD = 1063.3; CI 95% = 509.7–1616.9, *p* < 0.001) and group 3 (MD = 1682.3; CI 95% = 794.3–2570.3, *p* < 0.001), and SOD in group 2 (MD = 49.2; CI 95% = 29.1–69.2, *p* < 0.001) and group 3 (MD = 54.1; CI 95% = 27.4–80.8, *p* < 0.001). Meanwhile, serum levels of MDA in group 2 (MD = –0.4; CI 95% = −0.6 to −0.2, *p* < 0.001) and group 3 (MD = −0.3; CI 95% = −0.5 to −0.1, *p* = 0.004) significantly decreased. Moreover, no significant difference was observed in group 1 in terms of CAT, SOD, and MDA levels (*p* > 0.05) except for GPx level (MD= −52.3; CI 95% = −100.8 to –3.7, *p* = 0.036), which reduced remarkably (Table 4).

Before intervention, there were no significant differences between groups in terms of serum levels of insulin (*p* = 0.135) and HOMA-IR (*p* = 0.188) except for FBS (*p* = 0.003). After intervention, however, with adjusting baseline values, significant differences were observed in terms of FBS (*p* = 0.003), insulin (*p* = 0.029), and HOMA-IR (*p* = 0.017) levels. In the binary comparison, a significant decrease was seen in serum levels of FBS (aMD = −8.7; CI 95%= −14.7 to 2.7, *p* = 0.002) in group 3, insulin (aMD = −5.6; CI 95% = −10.8 to −0.4, *p* = 0.029) in group 2, and HOMA-IR (aMD = −1.3; CI 95% = −2.4 to −0.2, *p* = 0.013) in group 2 in comparison with group 1. Moreover, the binary comparison indicated no significant difference in the FBS level (*p* = 0.212) in group 2, the insulin level (*p* = 0.842) in group 3, and the HOMA-IR level (*p* = 0.403) in group 3 compared to group 1. Furthermore, based on our results from the binary comparison between groups 2 and 3 (*p* > 0.05), there was no significant difference in the FBS, insulin, and HOMA-IR levels.

Additionally, in within-group analysis, no significant decrease was detected after intervention compared to baseline regarding the serum levels of FBS (MD = −6.4; CI 95%= −10.5 to −2.3, *p* = 0.004) in group 3, insulin (MD = −4.3; CI 95% = −7.7 to −2.2, *p* < 0.001) in group 2, and HOMA-IR (MD = −1.0; CI 95% = −1.4 to −0.4, p = 0.002) in group 2; yet, in the within-group comparison, there were no significant differences in the FBS levels in groups 1 and 2, the insulin levels in groups 1 and 3, and the HOMA-IR levels in groups 1 and 3 (*p* > 0.05, Table 5).

Ultrasonography for PCOS women on the second day of menstrual cycle after three-month treatment demonstrated no significant variations in the number, size of basal antral follicle count (AFC), and volume of ovary in all three groups that were 10–12, 2–12 mm, and 10 cm^3^, respectively. In addition, sonography in the mid-cycle of the third month showed that in group 1, the antral follicles of 14 out of 20 patients reached the ovulation stage and 4 of them became pregnant (20%). While in group 2, 11 PCOS women out of 20 showed dominant follicles, 2 of which experienced the pregnancy (10%). In group 3, 17 out of 20 PCOS participants had dominant follicles (18–20 mm), which resulted in 5 pregnancies (25%; one of them was pregnant with twins). 

Our results also showed that after intervention, in groups 1, 2, and 3, 35%, 22.2%, and 35% had oligomenorrhea versus 65% (*p* = 0.014), 55.6% (*p* = 0.014), and 100% (*p* < 0.001) at the baseline. Furthermore, after intervention, 25% of women in group 1 had amenorrhea versus 45% at the baseline (*p* = 0.46), and 25% in group 3 had amenorrhea versus 65% at the baseline (*p* = 0.005). However, after intervention, in group 2, 16.7% had amenorrhea versus 33.3% at the baseline; though the difference between the groups was not significant (*p* = 0.083). Meanwhile, no side effects were reported in all three study groups during the 12 weeks of intervention.

## 4. Discussion

According to our results presented in Table 1, herbal mixture showed potent antioxidant activity, as well as total phenol and flavonoid content. Based on previous studies, the antioxidant activity of extracts could mainly be attributed to the polyphenolic compounds [34]. The FRAP assay is widely used to determine the antioxidant compounds in dietary polyphenols [35]. In fact, polyphenols determine the antioxidant activity, and this positive relationship renders a trend in many medicinal plants [36]. It has been proven that cinnamon contains a great variety of flavonoids and polyphenols with free-radical-scavenging properties and antioxidant activities [37]. Ginger with a wide range of antioxidants and polyphenol compounds such as β-carotene, ascorbic acid, terpenoids, alkaloids, and polyphenols like flavonoids, flavone glycosides, and rutin is regarded the greatest source of phytochemical antioxidants [38]. Phenolic phytochemicals of spearmint significantly enhance the antioxidant defense [39]. Peel of *C. sinensis* is also very rich in phenolic compounds, including flavonoids and phenolic acids, which can be consumed as natural antioxidants [40].

The GC–MS and GC–FID analyses in Table 2 showed the presence of zingiberene (13.58%), α-curcumene (10.77%), β-sesquiphellandrene (9.86%), α-farnesene (5.30%), and β-bisabolene (5.93%) with sesquiterpene hydrocarbons in ginger; cinnamic aldehyde (13.06%) with the organic compounds of aldehyde in cinnamon, and finally, pulegone (5.61%) with monoterpenes extracted from spearmint. A review of the literature indicated that sesquiterpene hydrocarbons and monoterpenes could be effective in various biological activities such as antioxidant effects [41,42]. Zhan et al. extracted α-zingiberene (22.29%), β-sesquiphellandrene (8.58%), α-farnesene (3.93%), β-bisabolene (3.87%), α-curcumene (2.63%) with sesquiterpene hydrocarbons from ginger volatile oil, which had antioxidant potency [43]. In addition, it was shown in another study that the antioxidant effects of essential oils of cinnamon barks and ginger rhizomes could be ascribed to the presence of antioxidant constituents such as cinnamaldehyde, sesquiphellandrene, and zingiberene [44]. Using the GC–MS analysis, Telci et al. determined the main composition of spearmint as oxygenated monoterpenes including pulegone (26.71–29.56%) and piperitone (22.17–28.16%) as the major terpenoid group [45]. Significant difference in the percentage of oil compositions in comparison to that of our study might depend on variation in environmental factors as well as difference in ecologies.

Regarding Table 4, our results clearly showed that after a 12-week intervention, levels of CAT, GPx, and SOD significantly increased in groups 2 and 3 compared to group 1. Although a significant decrease in case of MDA was seen in groups 2 and 3, there was no significant difference in comparison with group 1. Moreover, according to Table 5, levels of FBS in group 3, and insulin and HOMA-IR in group 2 significantly decreased compared to group 1.

Many studies have indicated low serum antioxidant levels and insulin resistance in PCOS women [46,47,48]. In fact, hyperglycemia can enhance OS through numerous pathways. However, through a major non-enzymatic mechanism, it induces the intracellular ROS, generates electrochemical proton gradient produced by mitochondrial electron transport chain (mETC), and results in enhanced derivation of superoxide [49]. In addition, reactive species by impairing glucose uptake in muscle and fat [50], and also by reducing insulin secretion from pancreatic β cells play a crucial role in insulin resistance [7]. On the other hand, an imbalance between oxidant-antioxidant is responsible for ovarian disturbance, insulin resistance, and chronic inflammation that is associated with pancreatic β cell dysfunction in women with PCOS [10]. Although ROS and RNS at low concentrations play physiological roles in cellular signaling pathways, in excess levels they may damage cellular functions [6]. Reactive oxygen species react with lipids, causing them to boost peroxidation products such as MDA [47].

Cinnamon also reduces insulin resistance by enhancing phosphatidylinositol 3-kinase activity in the insulin signaling pathway [51]. In several studies, the authors proved that cinnamon contained proanthocyanidins and phenolic compounds with antioxidant activities that not only neutralized free radicals, but also reduced blood glucose and insulin [52,53]. Moselhy et al. indicated a strong hepatoprotective effect of cinnamon ethanolic extract against carbon tetrachloride (CCl4)-induced oxidative stress by increasing the SOD and CAT levels and decreasing the MDA level, the possible mechanism of which could be attributed to the free radical scavenging activity of polyphenol compounds [54]. In a study examining the effect of ginger and onion (*Allium cepa*) on the levels of sexual hormones and OS, the reduction of MDA serum level was significant in the within-ginger group, but was not significant in comparison to the control group; this result was in line with ours [55]. A use of the combination of ginger and cinnamon in diabetic male rats in the study of Khaki et al. particularly illustrated an increase in the SOD, CAT, and GPx levels, while the MDA level more significantly decreased compared to other groups using only one herb [17]. In an animal study, Al-Amin et al. demonstrated that ginger could decrease the FBS level by the mechanism of serotonin receptors which activate pancreatic β cells to release insulin [56]. Additionally, Bayani et al. showed that phenolic phytochemicals of spearmint possessed hypoglycemic, and antioxidant attributes [57]. Deep et al. also revealed that methanolic extract of spearmint was a more potent antioxidant than ascorbic acid, which is recognized as the standard antioxidant. Furthermore, the levels of SOD, GPx, and CAT were duplicated, while MDA level fell by half [58]. In other research, Selmi et al. approved that hesperidin as one of the main flavonoids of *C. sinensis* reversed lipoperoxidation and hydrogen peroxide production, reduced MDA level, and increased antioxidant status (CAT, GPX, SOD levels) drastically [59]. 

Volatile oil from *C. sinensis* peel contains naringin and naringenin as two kinds of flavonoids which have anti-diabetic and antioxidant properties [60]. Polymethoxylated flavones (PMFs) of *C. sinensis* have hypolipidemic effects, resulting in a significant reduction of insulin tolerance and glucose levels [61]. Furthermore, according to other research, a mixture of onion, ginger, basil, cinnamon, orange peel, yellow and red watermelon seeds, and carrot seed could significantly affect the CAT level and reduce the OS [62]. All of this research is consistent with our study.

In this study, regular menstrual cycle, ovulation, and pregnancy occurred in all three groups. The main cause of this syndrome is the reduction in the sensitivity of pre-antral follicles to follicle stimulating hormone (FSH), and enhancement of follicular activity to luteinizing hormone (LH) leading to the inhibition of follicle maturation [63]. Similarly, excessive secretion of LH and gonadotropin-releasing hormone (GnRH) stimulate ovarian theca cells to produce androgens [64]. As a consequence, hyperandrogenism not only arrests antral follicle growth, but also stimulates apoptosis of its granulose cells which convert androgen to estradiol employing aromatase enzyme [65].

The study of Khodaeifar et al. certified that cinnamon could improve ovulatory menstrual cycle by enhancing progesterone levels in the luteal phase of PCOS women [66]. Similarly, Ataabadi et al. demonstrated that spearmint, through reducing body weight and level of testosterone and having antioxidant properties, matured follicles, and induced ovulation, led to higher number of Graafian follicles and corpus lutea and lower number of ovarian cysts and atretic follicles [24]. Ginger not only stimulated blood circulation for the treatment of inflammation and menstrual irregularities, but also enhanced AFC and ovarian reserve function [67]. Clomiphene citrate stimulated ovulation and increased fertility rate via stimulating the secretion of gonadotropin-releasing hormone as well as anti-estrogenic effect [68].

The results of the present study pointed out the fact that herbal mixture could increase the serum levels of CAT, GPx, SOD, and reduce the MDA level. Likewise, CC along with herbal mixture could exert such effects. This result may contribute to the point that although CC could not alter serum level of the antioxidant, it did not interfere in the activity of herbal supplement. Although the FBS level did not reduce in the groups administered with CC and herbal mixture solely, their combination could decrease the FBS; this result might be ascribed to the synergistic effects of their combination. Further studies are needed to prove these findings.

### Limitations of the Study

One of the major strengths of this study was not only the analysis of essential pharmacophores, TPC, TFC, DPPH, and FRAP in a herbal mixture, but also the evaluation of its efficacy on serum antioxidant levels and glycemic biomarkers in infertile PCOS women. Furthermore, in previous studies, the effects of all aforementioned herbs were investigated separately, while we evaluated the effects of CC and herbal mixture as a new combination, which might have synergistic effects. Nonetheless, the relatively small sample size, short-term follow-up, single-blind design, and using a fixed dose of herbal mixture in the clinical study were the limitations of this study.

We recommend that further research should be undertaken regarding the effects of herbal mixtures on hormonal factors of PCOS women such as sexual hormones in both proliferative and secretory phases, and lipid profile in a longer follow-up.

## 5. Conclusions

In summary, the results indicated that phenolics and phyto-antioxidant constituents of the herbal mixture including zingiberene, α-curcumene, β-sesquiphellandrene, α-farnesene, β-bisabolene, cinnamic aldehyde, and pulegone could effectively improve antioxidant potential, and the levels of FBS, HOMA-IR, and insulin in the serum of PCOS patients. It can be concluded that consumption of a herbal mixture as a supplement alongside CC can improve the antioxidant activity, glycemic status, and pregnancy rate in PCOS patients. Hence, it could be considered as a beneficial supplementary medicament for patients suffering from PCOS.

## Figures and Tables

**Figure 1 biomolecules-09-00215-f001:**
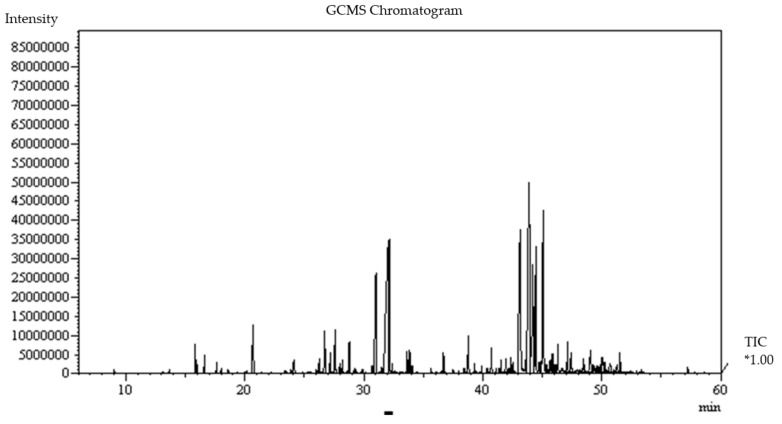
Gas chomatography–mass chromatogram (GC–MS) of the essential oil of herbal mixture. TIC*1.00: total ion current (1.0 std).

**Figure 2 biomolecules-09-00215-f002:**
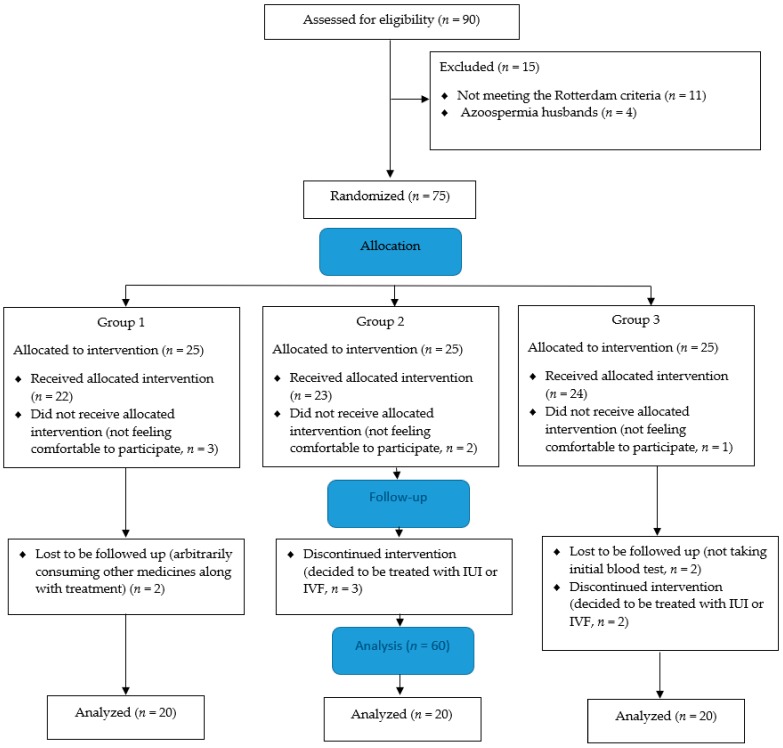
Polycystic ovary syndrome (PCOS) patients’ flow diagram. Group 1: Clomiphene citrate (CC), group 2: Herbal mixture, and group 3: CC along with herbal mixture. IUI: intrauterine insemination, IVF: in vitro fertilization.

**Table 1 biomolecules-09-00215-t001:** DPPH, FRAP, TPC, and TFC of methanolic extract of herbal mixture.

Test	DPPH (RC50) mg/mL	FRAP µmol Fe (II)/g	TPC mg Gallic acid/100 mg	TFC mg Quercetin/100 mg
Methanolic extract	0.018 ± 0.0007	720 ± 35	24.062 ± 0.2	8.93 ± 0.09
Quercetin	0.004 ± 0.0001	2880 ± 41	-	-

DPPH (2,2-diphenyl-1-picrylhydrazyl) free radical scavenging activity, FRAP: ferric reducing antioxidant potential, TPC: total phenolic content, TFC: total flavonoid content.

**Table 2 biomolecules-09-00215-t002:** Volatile compounds identified in the essential oil of herbal mixture.

No.	Rt.	% Area	Formula	KI	Peak
1	13.665	0.13	C_7_H_16_O	884	2-Heptanol
2	15.827	0.75	C_7_H_6_O	928	Benzaldehyde
3	15.946	0.30	C_10_H_16_	930	α-Pinene
4	16.607	0.41	C_10_H_16_	943	Camphene
5	17.628	0.24	C_8_H_14_O	962	6-Methyl-5-Hepten-2-One
6	18.009	0.10	C_10_H_16_	970	β-Pinene
7	18.563	0.10	C_8_H_18_O	980	3-Octanol
8	20.675	1.66	C_10_H_18_O	1020	Eucalyptol
9	24.119	0.50	C_10_H_18_O	1084	l-Linalool
10	26.022	0.12	C_10_H_16_O	1120	2-Camphonone
11	26.261	0.52	C_9_H_10_O	1125	Benzenepropanal
12	26.723	1.35	C_10_H_18_O	1134	Menthone
13	27.184	0.56	C10H18O	1143	Isomenthone
14	27.596	1.53	C_10_H_18_O	1150	Borneol
15	27.985	0.26	C_10_H_20_O	1158	Menthol
16	28.216	0.48	C_10_H_18_O	1162	4-Terpineol
17	28.788	0.98	C_10_H_18_O	1173	Linalyl propionate
18	30.716	0.25	C_10_H_20_O	1211	Citronellol
19	31.031	5.61	C_10_H_16_O	1217	Pulegone
20	31.502	0.21	C_10_H_16_O_2_	1227	P-Menthan-3-One, 1,2-Epoxy
21	32.048	13.06	C_9_H_8_O	1238	Cinnamic aldehyde
22	32.119	4.92	C_10_H_8_O_2_	1239	Benzalmalonic dialdehyde
23	32.38	0.30	C_10_H_16_O	1244	Citral
24	33.618	0.64	C_10_H_14_O	1269	Thymol
25	33.693	0.36	C_12_H_20_O_2_	1271	Bornyl acetate
26	33.858	0.59	C_11_H_22_O	1274	2-Undecanone
27	35.65	0.16	C_10_H_14_O	1311	Piperitenone
28	36.698	0.65	C_10_H_14_O_2_	1333	Piperitenone oxide
29	38.448	0.23	C_15_H_24_	1370	Cycloisosativene
30	38.797	1.39	C_15_H_24_	1378	Copaene
31	39.32	0.24	C_15_H_24_	1389	β-Elemene
32	40.733	0.79	C_15_H_24_	1420	Caryophyllene
33	41.339	0.13	C_15_H_24_	1434	α-Bergamotene
34	41.941	0.48	C_15_H_24_	1448	β-Farnesene
35	42.195	0.14	C_15_H_24_	1453	α-Humulene
36	42.526	0.24	C_15_H_24_	1461	Aromadendrene
37	43.15	10.77	C_15_H_22_	1475	α-Curcumene
38	43.914	13.58	C_15_H_24_	1492	Zingiberene
39	44.213	5.30	C_15_H_2_4	1499	α-Farnesene
40	44.483	5.93	C_15_H_24_	1505	β-Bisabolene
41	44.718	0.39	C_15_H_24_	1511	γ-Muurolene
42	44.859	0.36	C_15_H_22_	1514	Calamenene
43	45.117	9.86	C_15_H_24_	1520	β-Sesquiphellandrene
44	46.32	0.86	C_15_H_26_O	1549	Nerolidol
45	47.138	1.04	C_15_H_24_O	1569	Spathulenol
46	47.415	0.83	C_15_H_24_O	1575	Caryophyllene oxide
47	49.057	1.16	C_15_H_26_O	1615	Epiglobulol
48	49.217	0.28	C_15_H_26_O	1619	Cubenol
49	49.643	0.35	C_15_H_26_O	1630	α-Cadinol
50	49.742	0.16	C_15_H_26_O	1633	Torreyol
51	50.04	0.81	C_15_H_26_O	1640	B-Eudesmol
52	57.24	0.17	C_18_H_36_O	1830	Hexahydrofarnesyl Acetone

Monoterpenes and derivatives: 15.45. Sesquiterpenes and derivatives: 55.49. Aldehydes: 19.38. Other compounds: 1.91. Total identified: 92.23. KI: Kovats index, Rt: retention time.

**Table 3 biomolecules-09-00215-t003:** General characteristics of the women with PCOS at baseline and after intervention.

Maternal Data	Group 1 (*n* = 20)	Group 2 (*n* = 20)	Group 3 (*n* = 20)	*p*-Value
Age (year), mean ± SD 20–25, *n* (%) 26–30, *n* (%) 31–35, *n* (%)	25.0 ± 3.8 14 (70%) 4 (20%) 2 (10%	26.2 ± 4.4 8 (40%) 10 (50%) 2 (10%)	25.7 ± 4.2 12 (60%) 5 (25%) 3 (15%)	0.685 ^†^ 0.288 ^§^
Marital age (Year), mean ± SD	20.7 ± 3.7	20.1 ± 3.7	19.7± 2.9	0.689 ^†^
History of infertility (Year), mean ± SD	3.1 ± 1.4	4.3 ± 4.2	4.2 ± 2.7	0.374 ^†^
Weight (kg), mean ± SD (baseline)	72.8 ± 11.7	76.0 ± 11.0	72.6 ± 11.6	0.597 ^†^
BMI (kg/m^2^), mean ± SD (baseline)	27.1 ± 4.1	28.5 ± 3.4	26.9 ± 3.9	0.415 ^†^
BMI (kg/m^2^) mean ± SD (After intervention)	26.9 ± 0.2	27.0 ± 0.2	26.8 ± 0.2	0.770 ^¥^
Type of infertility *n* (%)				0.250 ^§^
Primary	18(90%)	16 (80%)	19(95%)	
Secondary	2(10%)	4 (20%)	1(5%)	
Smoking *n* (%)				
None	20 (100%)	20 (100%)	20 (100%)	

^†^ One-way variance analysis (ANOVA), ^§^ Fishers exact test, ^¥^ Analysis of covariance for between groups. Group 1: Clomiphene Citrate (CC), Group 2: herbal mixture, and Group 3: CC with herbal mixture (Clomiphene: Control). SD: standard deviation, BMI: body mass index.

**Table 4 biomolecules-09-00215-t004:** Comparison of mean ± SD serum levels of CAT, GPx, SOD, and MDA in participants receiving CC, herbal mixture, and CC along with herbal mixture.

Variable	Group 1		Group 2		Group 3		*p* _0_	*p* _1_	Adjusted MD (95% CI) ^1 × 2^	*p* ^2 × 1^	Adjusted MD (95% CI) ^1 × 3^	*p* ^3 × 1^	*p* ^3 × 2^
	Before	After	Before	After	Before	After							
CAT (pg/mL)	38.1 ± 7.0	38.9 ± 6.9	39.9 ± 8.0	48.5 ± 11.8	38.4 ± 6.5	51.2 ± 10.8	0.725	0.001	9.0 (1.1–16.9)	0.021	12.2 (4.5–19.9)	0.001	0.689
MD (95% CI)	0.7 (−0.0 to 1.5) *		8.6 (1.9–15.2) *		12.8 (6.5–19.1) *								
*p* ^†^	0.077		0.014		<0.001								
GPx (IU/mL)	7375.0 ± 1647.3	7322.7 ± 1664.1	7119.4 ± 930.8	8281.8 ± 767.4	7466.9 ± 1619.9	9149.3 ± 1459.8	0.751	<0.001	986.1 (141.1–1831.1)	0.017	1781.2 (960.7–2601.8)	<0.001	0.072
MD (95% CI)	−52.3 (−100.8 to −3.7) *		1063.3 (509.7–1616.9) *		1682.3 (794.3–2570.3) *								
*p* ^†^	0.036		0.001		0.001								
SOD (IU/mL)	212.7 ± 21.4	208.7 ± 20.2	215.2 ± 28.3	264.4 ± 22.1	210.0 ± 20.1	264.1 ± 54.4	0.793	<0.001	55.1 (26.0–84.2)	<0.001	55.9 (27.5–84.2)	<0.001	1.0
MD (95% CI)	−3.9 (−8.1 to 0.2) *		49.2 (29.1–69.2) *		54.1 (27.4–80.8) *								
*p* ^†^	0.063		<0.001		<0.001								
MDA (µMol/L)	1.7 ± 0.2	1.5 ± 0.3	1.8 ± 0.3	1.4 ± 0.3	1.8 ± 0.3	1.4 ± 0.4	0.835	0.420	−0.1 (−0.5 to 0.1)	0.502	−0.1 (−0.4 to 0.2)	0.725	0.978
MD (95% CI)	−0.1 (−0.3 to −0.1) *		−0.4 (−0.6 to −0.2) *		−0.3 (−0.5 to −0.1) *								
*p* ^†^	0.052		0.001		0.004								

CAT: Catalase, GPx: Glutathione peroxidase, SOD: Superoxide dismutase, MDA: Malondialdehyde. *p*
^2 × 1^ Difference between groups 2 and 1 after intervention, *p*
^3 × 1^ Difference between group 3 and 1 after intervention, *p*
^3 × 2^ Difference between groups 3 and 2 after intervention. *p*
^†^: Paired-Samples *t*-test for within-group comparison, *p*_0_: One-way ANOVA for between-group comparison at baseline, *p*_1_: Analysis of covariance for between-group comparison after intervention adjusted for baseline, * 95% CI within-group comparison. All numbers are given as mean ± SD except for those specified as mean differences (MD) or *p*-value. Group 1: Clomiphene Citrate (CC), Group 2: herbal mixture, and Group 3: CC with herbal mixture (Clomiphene: Control).

**Table 5 biomolecules-09-00215-t005:** Comparison of mean ± SD serum levels of FBS, insulin, and HOMA-IR in participants receiving CC, herbal mixture, CC along with herbal mixture.

Variable	Group 1		Group 2		Group 3		*p* _0_	*p* _1_	Adjusted MD (95% CI) ^2×1^	*p* ^2 × 1^	Adjusted MD (95% CI) ^3×1^	*p* ^3 × 1^	*p* ^3 × 2^
	Before	After	Before	After	Before	After							
FBS (mg/dL)	95.1 ± 6.4	92.3 ± 2.8	85.9 ± 11.8	83.5 ± 8.8	85.5 ± 9.5	79.0 ± 8.0	0.003	0.003	−4.4 (−10.5 to 1.6)	0.212	−8.7 (−14.7 to −2.7)	0.002	0.195
MD (95% CI)	−2.8 (−5.6 to −0.0) *		−2.4 (−7.7 to 2.8) *		−6.4 (−10.5 to −2.3) *								
*p* ^†^	0.152		0.344		0.004								
Insulin (µIU/mL)	17.1 ± 7.2	17.8 ± 1.4	16.4 ± 5.5	12.1 ± 1.5	20.8 ± 8.5	16.2 ± 1.4	0.135	0.029	−5.6 (−10.8 to −0.4)	0.029	−1.6 (−6.7 to 3.6)	0.842	0.182
MD (95% CI)	0.7 (−2.2 to 2.9) *		−4.3 (−7.7 to −2.2) *		−4.6 (−8.4 to 1.3) *								
*p* ^†^	0.852		0.001		0.156								
HOMA-IR	4 ± 1.6	3.9 ± 1.8	3.4 ± 1.1	2.3 ± 0.3	4.4 ± 1.9	3.4 ± 1.7	0.188	0.017	−1.3 (−2.4 to −0.2)	0.013	−0.6 (−1.6 to 0.4)	0.403	0.334
MD (95% CI)	−0.1 (−0.6 to 0.4) *		−1.0 (−1.7 to −0.4) *		−0.9 (−1.9 to 0.0) *								
*p* ^†^	0.715		0.002		0.065								

FBS: Fast blood sugar, HOMA-IR: Homeostatic model assessment for insulin resistance, *p*
^2 × 1^: Difference between groups 2 and 1 after intervention, *p*
^3 × 1^: Difference between groups 3 and 1 after intervention, *p*
^3 × 2^: Difference between groups 3 and 2 after intervention. *p*
^†^: Paired-Samples *t*-Test for within-group comparison, *p*_0_: One-way ANOVA for between-group comparison at baseline, *p*_1_: Analysis of covariance for between-group comparison after intervention adjusted for baseline, * 95% CI within-group comparison. All numbers are given as mean ± SD except for those specified as mean differences (MD) or *p*-value. Group 1: Clomiphene Citrate (CC), Group 2: Herbal mixture, and Group 3: CC with herbal mixture (Clomiphene: Control).

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
