# Peer review of "The Effectiveness of Herbal Mixture Supplements with and without Clomiphene Citrate in Comparison to Clomiphene Citrate on Serum Antioxidants and Glycemic Biomarkers in Women with Polycystic Ovary Syndrome Willing to Be Pregnant: A Randomized Clinical Trial"

_biomolecules, 2019, doi:10.3390/biom9060215_

Reviewer 1 Report

Dear Author,

This work is a single-blind randomized clinical trial that aimed to evaluate the effect of an herbal mixture supplement on serum antioxidants and insulin-resistance indexes in women treated or not with clomiphene citrate.

Of note, HOMA and insulin are not glycaemic indexes (this term has a different meaning), but HOMA is an index of insulin resistance.

The Introduction is clear, with adequate bibliographical references, and summarizes the scientific background and the aims of the authors. The design of the study is well defined.

However, the methods and the results should be differently structured. The analysis and preparation of the plant mix and the standardization of the herbs should precede the clinical part, related to the trial, randomization and analysis of the levels of MDA, SOD, CAT, GPx, insulin in women.

It should be better to avoid excessive division of materials and methods into many subparagraphs.

The results include the same data entered in the tables: this is redundant, only a source of results could be left. In Table 1 the data should be expressed as mean+DS or n(%) and this information should be described near each variable (not in the notes below).

The table 2 and Figure 2 show the same data, so they are redundant. Consider to insert one of them in the supplements.

Moreover, in Results it is conceptually correct to start with the data of preparation and analysis of the herbal mixture, and then to describe the analysis of women.

Binary analysis between group 2 and group 3 is missing but it could give additional and interesting information.

The Discussion is wide, but only with regard to antioxidant activity of herbal mix. References and discussion about the effects on PCOS women are missing.

Moreover, although the limitations of the study have been taken into consideration in the Discussion, the preliminary role of data is not sufficiently underlined. The lack of association with clinical data makes the scientific relevance of the study quite poor.

Author Response

Thanks a lot for valuable comments of editors and reviewers on manuscript entitled: " The effectiveness of herbal mixture supplements with and without clomiphene citrate in comparison to clomiphene citrate on serum antioxidants and glycemic biomarkers in women with polycystic ovary syndrome willing to be pregnant: a randomized clinical trial" that helped us to improve our manuscript. The requested correction was completely done in the manuscript as below:

Authors' comments

Of note, HOMA and insulin are not glycaemic indexes (this term has a different meaning), but HOMA is an index of insulin resistance.

R: Glycemic indexes converted to glycemic biomarkers/ status.

The Introduction is clear, with adequate bibliographical references, and summarizes the scientific background and the aims of the authors. The design of the study is well defined.

- However, the methods and the results should be differently structured. The analysis and preparation of the plant mix and the standardization of the herbs should precede the clinical part, related to the trial, randomization and analysis of the levels of MDA, SOD, CAT, GPx, insulin in women.

R: The methods and the results structured as you mentioned.

It should be better to avoid excessive division of materials and methods into many subparagraphs.

R: Subparagraphs of materials and methods were merged.

The results include the same data entered in the tables: this is redundant, only a source of results could be left. In Table 1 the data should be expressed as mean+DS or n(%) and this information should be described near each variable (not in the notes below).

R: All of requests were performed.

The table 2 and Figure 2 show the same data, so they are redundant. Consider to insert one of them in the supplements.

R: Since Table 2 provide more comprehensive data, we omitted Figure 2.

Moreover, in Results it is conceptually correct to start with the data of preparation and analysis of the herbal mixture, and then to describe the analysis of women.

R: The results re-structured as you mentioned.

Binary analysis between group 2 and group 3 is missing but it could give additional and interesting information.

R: Binary analysis between group 2 and group 3 were mentioned in both tables and result section.

The Discussion is wide, but only with regard to antioxidant activity of herbal mix. References and discussion about the effects on PCOS women are missing.

R: Explanations about antioxidant activity were summarized and discussion about the effects of intervention on PCOS women was added.

Moreover, although the limitations of the study have been taken into consideration in the Discussion, the preliminary role of data is not sufficiently underlined. The lack of association with clinical data makes the scientific relevance of the study quite poor.

R: According toIRCT201509295563N7, we had some clinical data and regarding to reviewer's comment we included them in the result of clinical trial section.

==============================================================================================

Include the ethical approval number under the 'Clinical trial study' section.

R: It is done.

The authors may include more details on the methodology.

R: We included more details on the methodology

The basis of selection ratio 5:4:3:2 may be detailed.

R: The basis of selection ratio 5:4:3:2 explained in the result.

Results: The entire results section needs to be presented as Mean ± SD/SEM.

R: All results section was presented as Mean ± SD/SEM.

The authors need to apply Mann–Whitney U test.

R: Normality of all quantitative variables for each of the groups was re-investigated by Kolmogorov-Smirnov test and confirmed. Since all mentioned variable had normal distribution, we used parametric tests.

The chromatographic conditions needs to be explained and needs to be presented in the results section.

R: It is done.

The authors need to include ABTS radical scavenging activity along with DPPH assay.

R: In order to do ABTS assay, I reached out to numerous laboratories in Tabriz to help me perform it, unfortunately they don't have ABTS material available. They informed me that due to sanction, they are not able order or import laboratory material like they used to in the past.

I explored other option from google scholar to see if any university in Iran had previously done this assay. I found some articles published in 2018-2019 from Tehran and Shiraz University that they did this experiment, I reached out to the corresponds to see if I could fly there and use their lab, but unfortunately they also said they did this research 2-3 years ago and published their article last year, and they don't have this material available either. At this point my hands are tight, and I performed FRAP test instead.

The base peak chromatograms needs to be added under the results section.

R: It is added under the results section

Thank you

Reviewer 2 Report

Include the ethical approval number under the 'Clinical trial study' section.

The authors may include more details on the methodology.

The basis of selection ratio 5:4:3:2 may be detailed.

Results: The ntire results section needs to be presented as Mean ± SD/SEM.

The authors need to apply Mann–Whitney U test.

The chromatographic conditions needs to be explained and needs to be presented in teh results section.

The authors need to include ABTS radical scavenging activity along with DPPH assay.

The base peak chromatograms needs to be added under the results section.

Author Response

Thanks a lot for valuable comments of editors and reviewers on manuscript entitled: " The effectiveness of herbal mixture supplements with and without clomiphene citrate in comparison to clomiphene citrate on serum antioxidants and glycemic biomarkers in women with polycystic ovary syndrome willing to be pregnant: a randomized clinical trial" that helped us to improve our manuscript. The requested correction was completely done in the manuscript as below:

Authors' comments

Of note, HOMA and insulin are not glycaemic indexes (this term has a different meaning), but HOMA is an index of insulin resistance.

R: Glycemic indexes converted to glycemic biomarkers/ status.

The Introduction is clear, with adequate bibliographical references, and summarizes the scientific background and the aims of the authors. The design of the study is well defined.

- However, the methods and the results should be differently structured. The analysis and preparation of the plant mix and the standardization of the herbs should precede the clinical part, related to the trial, randomization and analysis of the levels of MDA, SOD, CAT, GPx, insulin in women.

R: The methods and the results structured as you mentioned.

It should be better to avoid excessive division of materials and methods into many subparagraphs.

R: Subparagraphs of materials and methods were merged.

The results include the same data entered in the tables: this is redundant, only a source of results could be left. In Table 1 the data should be expressed as mean+DS or n(%) and this information should be described near each variable (not in the notes below).

R: All of requests were performed.

The table 2 and Figure 2 show the same data, so they are redundant. Consider to insert one of them in the supplements.

R: Since Table 2 provide more comprehensive data, we omitted Figure 2.

Moreover, in Results it is conceptually correct to start with the data of preparation and analysis of the herbal mixture, and then to describe the analysis of women.

R: The results re-structured as you mentioned.

Binary analysis between group 2 and group 3 is missing but it could give additional and interesting information.

R: Binary analysis between group 2 and group 3 were mentioned in both tables and result section.

The Discussion is wide, but only with regard to antioxidant activity of herbal mix. References and discussion about the effects on PCOS women are missing.

R: Explanations about antioxidant activity were summarized and discussion about the effects of intervention on PCOS women was added.

Moreover, although the limitations of the study have been taken into consideration in the Discussion, the preliminary role of data is not sufficiently underlined. The lack of association with clinical data makes the scientific relevance of the study quite poor.

R: According toIRCT201509295563N7, we had some clinical data and regarding to reviewer's comment we included them in the result of clinical trial section.

==============================================================================================

Include the ethical approval number under the 'Clinical trial study' section.

R: It is done.

The authors may include more details on the methodology.

R: We included more details on the methodology

The basis of selection ratio 5:4:3:2 may be detailed.

R: The basis of selection ratio 5:4:3:2 explained in the result.

Results: The entire results section needs to be presented as Mean ± SD/SEM.

R: All results section was presented as Mean ± SD/SEM.

The authors need to apply Mann–Whitney U test.

R: Normality of all quantitative variables for each of the groups was re-investigated by Kolmogorov-Smirnov test and confirmed. Since all mentioned variable had normal distribution, we used parametric tests.

The chromatographic conditions needs to be explained and needs to be presented in the results section.

R: It is done.

The authors need to include ABTS radical scavenging activity along with DPPH assay.

R: In order to do ABTS assay, I reached out to numerous laboratories in Tabriz to help me perform it, unfortunately they don't have ABTS material available. They informed me that due to sanction, they are not able order or import laboratory material like they used to in the past.

I explored other option from google scholar to see if any university in Iran had previously done this assay. I found some articles published in 2018-2019 from Tehran and Shiraz University that they did this experiment, I reached out to the corresponds to see if I could fly there and use their lab, but unfortunately they also said they did this research 2-3 years ago and published their article last year, and they don't have this material available either. At this point my hands are tight, and I performed FRAP test instead.

The base peak chromatograms needs to be added under the results section.

R: It is added under the results section

Thank you

Round  2

Reviewer 1 Report

Daer Authors,

The work has been sufficiently modified, while remaining in my opinion of not high scientific relevance.

Kind regards.

Author Response

Minor comments

The authors has to critically read the manuscript for English language. 

R: It is edited by scientific language editor. All revisions are clearly highlighted by Track Changes.

The references need to be cross-checked with the statements for authenticity

R: All references are cross-checked and sorted based on Endnote style of the journal.

Reviewer 2 Report

The authors addressed all the reviewer queries.

Minor comments

The authors has to critically read the manuscript for English language.

The references need to be cross-checked with the statements for authenticity

Author Response

(The authors gave the same response as above.)
